# Epigenetic and genetic components of height regulation

Stefania Benonisdottir[1,*], Asmundur Oddsson[1,*], Agnar Helgason[1,2], Ragnar P. Kristjansson[1], Gardar Sveinbjornsson[1], Arna Oskarsdottir[1], Gudmar Thorleifsson[1], Olafur B. Davidsson[1], Gudny A. Arnadottir[1], Gerald Sulem[1], Brynjar O. Jensson[1], Hilma Holm[1], Kristjan F. Alexandersson[1], Laufey Tryggvadottir[3,4], G. Bragi Walters[1], Sigurjon A. Gudjonsson[1], Lucas D. Ward[1], Jon K. Sigurdsson[1], Paul D. Iordache[1,5], Michael L. Frigge[1], Thorunn Rafnar[1], Augustine Kong[1,6], Gisli Masson[1], Hannes Helgason[1,6], Unnur Thorsteinsdottir[1,3], Daniel F. Gudbjartsson[1,6], Patrick Sulem[1] & Kari Stefansson[1,3]

Adult height is a highly heritable trait. Here we identified 31.6 million sequence variants by whole-genome sequencing of 8,453 Icelanders and tested them for association with adult height by imputing them into 88,835 Icelanders. Here we discovered 13 novel height associations by testing four different models including parent-of-origin ($|\beta| = 0.4$–10.6 cm). The minor alleles of three parent-of-origin signals associate with less height only when inherited from the father and are located within imprinted regions (IGF2-H19 and DLK1-MEG3). We also examined the association of these sequence variants in a set of 12,645 Icelanders with birth length measurements. Two of the novel variants, (IGF2-H19 and TET1), show significant association with both adult height and birth length, indicating a role in early growth regulation. Among the parent-of-origin signals, we observed opposing parental effects raising questions about underlying mechanisms. These findings demonstrate that common variations affect human growth by parental imprinting.

[1] deCODE Genetics/Amgen, Inc., 101 Reykjavik, Iceland. [2] Department of Anthropology, University of Iceland, 101 Reykjavik, Iceland. [3] Faculty of Medicine, University of Iceland, 101 Reykjavik, Iceland. [4] Icelandic Cancer Registry, 105 Reykjavik, Iceland. [5] Reykjavik University, 101 Reykjavik, Iceland. [6] School of Engineering and Natural Sciences, University of Iceland, 107 Reykjavik, Iceland. * These authors contributed equally to this work. Correspondence and requests for materials should be addressed to P.S. (email: patrick.sulem@decode.is) or to K.S. (email: kstefans@decode.is).

Height is an easily assessed trait in humans, with a narrow-sense heritability estimated to be 80% (refs 1–4). Abnormal stature is one of the most common phenotypic features of Mendelian conditions, as catalogued in the Online Mendelian Inheritance in Man (OMIM) (Supplementary Table 1). A large genome-wide association study (GWAS) meta-analysis by the GIANT consortium, based on the imputation of 2.6 million sequence variants into 253,288 individuals of European ancestry (including 26,799 Icelanders), discovered 697 sequence variants associated with height. Most are common (Minor allele frequency (MAF) > 5%) and together they explain 16% of the variance in height[5].

Previously, we performed GWAS for various diseases and other traits based on variants identified through whole-genome sequencing (WGS) of Icelanders (median read depth of 20X) and subsequent imputation into individuals from the same population. This led to the uncovering of associations of many low frequency and rare sequence variants with a large number of traits[6–12]. Discovery of uncommon and rare variants through GWAS is facilitated by founder events and the allelic homogeneity of the Icelandic gene pool, resulting from a relatively small population size during the past 1,100 years[13]. Moreover, extensive genealogical records enable the determination of the parent-of-origin of each genotyped or imputed allele, making it possible to evaluate the phenotypic impact of maternal and paternal inheritance of any given allele[14]. Phenotypic differences that result from the parental origin of alleles are usually attributed to epigenetic processes. The best known is genomic imprinting, wherein gene dosage is affected by the monoallelic expression of a particular gene, due to local epigenetic silencing of transcription from one chromosome[15]. In addition, the full genotypes of parents can sometimes influence the phenotype of offspring, for example through in utero effects from the mother. Such parental genotype effects can be hard to distinguish from parent-of-origin effects arising from genomic imprinting[15]. Genomic imprinting is estimated to affect about 1% of the genes in the human genome[16,17] (Supplementary Data 1). These genes typically occur in large clusters, where the non-random monoallelic expression depending on parental origin is maintained by imprinting control regions (ICR), that are differentially methylated regions (DMR) depending on parental origin. Parental specific methylation of DMRs is acquired in germ cells through epigenetic reprogramming and persists into adulthood[18,19]. The importance of ICRs for the regulation of gene dosage is revealed by the study of imprinting disorders. The best known example is provided by the growth factor IGF2 and the closely linked long noncoding RNA (lncRNA) growth suppressor H19, which are reciprocally imprinted genes located on the distal end of chromosome 11 that have been intensively studied both as a model system for understanding mechanisms of genomic imprinting and growth[20]. The dysregulation of IGF2 gene expression is associated with the growth disorders Beckwith–Wiedemann syndrome and Silver–Russell dwarfism (OMIM, http://www.omim.org; accessed 17.02.2016)[20,21].

Parent-of-origin effects and genomic imprinting have been shown to contribute to variation in complex traits in both animal models and humans[22–24]. As the vast majority of GWAS cannot assign parent-of-origin to an allele, there have been few reported GWASs of complex traits that apply parent-of-origin specific models[25–28], and we are aware of only one such association with height[29].

Adult height is to some extent determined by factors in utero[30,31] and numerous imprinted genes are essential for regulating fetal growth and placental development[32–34]. It is therefore of interest to assess the effect of adult height variants on birth length.

Here, we search for sequence variants associated with adult height in the Icelandic population and their effect on birth length using parent-of-origin, additive and recessive models.

## Results

**Study design.** A total of 31.6 million high quality sequence variants (84% single nucleotide variants and 16% short indels) with imputation information > 0.8 were detected through high coverage WGS of 8,453 Icelanders (median depth of 32X in genome build hg38). We imputed the variants into 150,656 individuals genotyped on Illumina microarray single-nucleotide polymorphism (SNP) chips, and 294,212 of their first and second-degree relatives. In total, 88,835 Icelanders with adult height measurements were used in the association study (average and s.d. for year of birth is 1948 ± 19 years, and for the height of males and females 178.8 ± 6.9 cm$^2$ and 165.6 ± 6.3 cm$^2$, respectively. The overall s.d. used in the paper is 6.6 cm) (Supplementary Fig. 1 and Supplementary Table 2). The imputed variants were tested for association with height under parent-of-origin models, in addition to additive and recessive models as previously described[14]. The threshold for genome-wide significance was corrected for multiple testing using a class-specific Bonferroni procedure based on functional impact of classes of variants[35] (Methods and Supplementary Fig. 2). We assessed the effect of the adult height variants on birth length in a set of 12,645 individuals (year of birth = 1989 ± 4 years, and the birth length of males and females was 52.1 ± 2.5 cm and 51.3 ± 2.5 cm, respectively). Only 539 of these individuals also had adult height measurements.

**Novel height associated signals.** We found four signals associating with adult height using parent-of-origin models, 58 under the additive model and one under the recessive model (Supplementary Data 2–5). Of these 63 association findings, 13 are novel, including three parent-of-origin signals, nine additive signals (mostly rare and low frequency coding variants) and one recessive signal (Table 1). The results for the novel additive and recessive signals are discussed in more detail in Supplementary Discussion (Supplementary Figs 3–12; Supplementary Tables 3 and 4) whereas the focus of the main text is on the parent-of-origin signals. A further 12 variants represent refinements of previously known signals (Supplementary Data 4) that together with the novel signals explain 2.1% of the variance of height in Iceland, on top of 16% attributable to previously reported association signals[5] (Supplementary Table 5, Methods).

**Parent-of-origin models.** The four association signals with parent-of-origin effects are all located in known imprinted regions (Table 2; Supplementary Data 1, Methods). The three novel signals, two at IGF2-H19 and one at DLK1-MEG3 imprinted loci, contain minor alleles that reduce height when paternally inherited (Table 2). The fourth was a confirmation of a signal in the KCNQ1 gene, recently described in the Sardinian population[29], where the minor allele reduces height when maternally inherited. We note that none of these four variants are correlated with seven variants previously reported with parent-of-origin association for other traits ($r^2 < 0.036$) (refs 14,27) (Supplementary Table 6).

Three of the four parent-of-origin height variants, two at IGF2-H19 and one at KCNQ1 loci, are located within a well-studied imprinted region of ∼1.8 Mb on chromosome 11p15 (GeneImprint, http://www.geneimprint.com; accessed 10.10.2015) (ref. 21). The regulation of imprinted genes in this region can be divided into two domains, each controlled in cis by its own ICR that maintains parental-specific monoallelic expression[21]. Two of the variants (rs147239461 and rs7482510) are located in the more telomeric of the two domains that also harbours the paternally expressed growth factor IGF2 gene and the maternally expressed lncRNA H19[21] (Fig. 1). Despite being only 200 Kb apart, these variants are not correlated ($r^2 = 0.014$)

**Table 1 | Summary information for variants representing novel associations with height.**

| rs name | Position (hg38) | $P$ value | $\beta$ (s.d.) | MAF (%) | Gene/ locus | Minor/ major | Impact | HGVS | Category | Dist (Kb) | OMIM |
|---|---|---|---|---|---|---|---|---|---|---|---|
| *Paternal model* | | | | | | | | | | | |
| rs147239461 | chr11:1,965,172 | $5.9 \times 10^{-13}$ | $-0.12$ | 5.09 | *IGF2-H19* | T/G | intergen | – | LOW | 36 | NC |
| rs7482510 | chr11:2,169,361 | $5.1 \times 10^{-11}$ | $-0.07$ | 16.84 | *IGF2-H19* | G/C | intron | – | LOW | 28 | NC |
| rs41286560 | chr14:100,883,117 | $2.2 \times 10^{-8}$ | $-0.12$ | 3.20 | *RTL1* | T/G | msns | NP_001128360.1: p.Pro558Thr | MOD | 1,554 | — |
| | | | | | | | | | | | |
| *Recessive model* | | | | | | | | | | | |
| rs62623707 | chr5:135,952,943 | $2.0 \times 10^{-10}$ | $-0.37$ | 5.56 | *LECT2* | G/A | msns | NP_002293.2: p.Ile24Thr | MOD | 750 | — |
| | | | | | | | | | | | |
| *Additive model* | | | | | | | | | | | |
| – | chr1:150,553,749 | $2.0 \times 10^{-8}$ | $-0.20$ | 0.79 | *ADAMTSL4* | C/CCAGAG CCCAGGC CTCTGGCA | fs | NP_001275536.1: p.Gln256ProfsTer38 | LOF | 340 | — |
| rs201828593 | chr8:134,610,589 | $8.0 \times 10^{-8}$ | 0.20 | 0.69 | *ZFAT* | G/C | msns | NP_001025110.2: p.Arg160Thr | MOD | 31 | — |
| rs75596750 | chr8:134,610,608 | $8.9 \times 10^{-10}$ | 0.32 | 0.35 | *ZFAT* | A/G | msns | NP_001025110.2: p.Arg154Trp | MOD | 31 | — |
| – | chr9:35,807,109 | $3.6 \times 10^{-31}$ | $-1.53$ | 0.058 | *NPR2* | T/G | msns | NP_003986.2: p.Gly869Val | MOD | 1 | 602875 |
| rs558226101 | chr10:68,686,650 | $1.6 \times 10^{-8}$ | 0.48 | 0.13 | *TET1* | T/C | msns | NP_085128.2: p.Arg1783Trp | MOD | 580 | — |
| – | chr15:88,872,939 | $1.2 \times 10^{-9}$ | $-1.61$ | 0.014 | *ACAN* | A/G | msns | NP_037359.3: p.Cys2416Tyr | MOD | 59 | 608361, 608361, 165800, 612813 |
| rs72755233 | chr15:100,152,748 | $8.7 \times 10^{-27}$ | $-0.10$ | 13.86 | *ADAMTS17* | A/G | msns | NP_620688.2: p.Thr446Ile | MOD | 5 | 613195 |
| – | chr17:63,918,372 | $1.0 \times 10^{-15}$ | $-0.26$ | 0.93 | *GH1* | C/G | msns | NP_000506.2: p.Leu49Val | MOD | 12 | 173100, 262400, 262650, 612781 |
| rs62621197 | chr19:8,605,262 | $7.3 \times 10^{-14}$ | $-0.16$ | 2.40 | *ADAMTS10* | T/C | msns | NP_112219.3: p.Arg62Gln | MOD | 26 | 277600 |

$\beta$(s.d.): effect in s.d. units (1 s.d._male = 6.9 cm, 1 s.d._female = 6.3 cm). MAF(%): Minor allele frequency in percentages. Impact: divided into fs: framshift, msns: missense, intron:intronic or intergen:intergenic. Category: the impact of each variant falls into one of these three categories, LOF: loss-of-funciton, MOD: moderate impact, LOW: Low impact. Dist: physical distance to the closest variant reported by GIANT. OMIM: accession number of disease with abnormality of height as a phenotypic feature linked to genes harbouring novel coding variants associated with height. Loci of non-coding variants (NC) are not included. A chi-squared test was used to calculate $P$-values.

and conditional analysis shows that they are independently associated with height (Supplementary Table 7).

The former, rs147239461, has nine strong correlates ($r^2 > 0.8$) (Fig. 2; Supplementary Fig. 13), clustering within 65 Kb and are located within 30 Kb downstream of *H19* at the 3′ boundary of the *IGF2-H19* imprinted domain (Fig. 1, Supplementary Data 6). The minor allele of rs147239461[T] is associated with reduced height when paternally inherited (MAF = 5.09 %, $\beta_{pat} = -0.12$ s.d. (corresponding to $-0.8$ cm), $P_{pat} = 5.9 \times 10^{-13}$; a chi-squared test was used to calculate $P$-values for all GWAS associations.). In contrast, when maternally inherited, the same allele is associated with greater height ($\beta_{mat} = 0.056$ s.d. (0.4 cm), $P_{mat} = 9.4 \times 10^{-4}$). The difference between the paternal and maternal effect of rs147239461[T] is significant ($P_{pat\,versus\,mat} = 1.2 \times 10^{-13}$) (Table 2). When parent-of-origin models are applied, two heterozygous genotypes of a biallelic marker are generated depending on parent of origin of the alleles. A comparison of mean height across all four possible genotype groups for rs147239461 revealed the greatest difference (1.1 cm, 0.16 s.d.) between the two different heterozygotes (Fig. 3). We will refer to this signal as IGF2-H19(A). Out of the nine strong correlates of rs147239461, three are located within binding sites for the highly conserved DNA-binding protein CTCF that can act as an insulator by blocking interactions between enhancers and promoters[36] frequently involved in imprinting regulation[37] (Supplementary Data 6). Also of note, the correlated variant rs75676658 which is 9 Kb from the H19-ICR is located in a muscle-specific promoter, a promotor type that constitutes 0.28% of the genome (Methods), which is interesting in the light that *IGF2* has been shown to promote mesoderm formation in mammals[38].

The second signal in the telomeric imprinted domain of 11p15 is represented by rs7482510, which is located 28 Kb upstream of *IGF2* (Fig. 1). It has no strong correlates according to either the deCODE data or the 1,000 Genomes phase 1 data (max $r^2 = 0.66$; SNP SNAP, https://www.broadinstitute.org/mpg/snap/ldsearch.php; accessed 09.02.2016). The minor allele G associates with less height when paternally inherited (MAF = 16.84 %, $\beta_{pat} = -0.065$ s.d. ($-0.4$ cm), $P_{pat} = 5.1 \times 10^{-11}$) (Table 2, Fig. 2, Supplementary Fig. 14), but does not affect height when maternally inherited ($\beta_{mat} = 0.018$ s.d. (0.1 cm), $P_{mat} = 0.076$) (Table 2). The variant rs7482510 is located at an oestrogen receptor (ESR1) binding site as observed in ENCODE ChIP-seq data (Supplementary Data 6). Interestingly, ESR1 is the main oestrogen receptor regulating skeletal growth[39]. We will refer to this signal as *IGF2-H19(B)*.

The third parent-of-origin association signal at 11p15 is best captured by rs143840904[T] when maternally inherited ($\beta_{mat} = -0.26$ s.d. ($-1.7$ cm), $P_{mat} = 2.0 \times 10^{-17}$). The SNP rs143840904[T] is located in the more centromeric domain at 11p15 in an intron 15 of *KCNQ1*. This signal was first described in the Sardinian population and was there represented by six highly correlated variants, ($r^2 > 0.70$), including rs143840904, that were all at similar frequency (MAF_Sardinia = 7.60 − 10.60%, $P_{Sardinia} = 5.2 \times 10^{-7}$–$5.6 \times 10^{-9}$) (ref. 29). We were able to both confirm the association and refine it to rs143840904[T], despite a lower frequency (MAF_Iceland = 1.78% versus MAF_Sardinia = 9.40%) (Supplementary Data 3). In contrast to the situation in Sardinia, rs143840904 has no strong correlates in Iceland (all $r^2 < 0.73$) (Fig. 2 and Supplementary Fig. 15), and conditional analysis revealed that it alone accounts for the maternal effect ($P_{adj} > 0.05$ for each of the other five variants,

**Table 2 | Parent-of-origin associations with height.**

| SNV ID | Position(hg38) | Minor/ major | MAF (%) | Impact | Locus | Additive | | Paternal | | Maternal | | Pat. versus Mat. |
|---|---|---|---|---|---|---|---|---|---|---|---|---|
| | | | | | | P | β (s.d.) | P | β (s.d.) | P | β (s.d.) | P |
| rs147239461 | chr11:1,965,172 | T/G | 5.09 | intergen | IGF2-H19 | 0.0028 | −0.043 | $5.9 \times 10^{-13}$ | −0.12 | $9.4 \times 10^{-4}$ | 0.056 | $1.2 \times 10^{-13}$ |
| rs7482510 | chr11:2,169,361 | G/C | 16.84 | intron | IGF2-H19 | $4.5 \times 10^{-4}$ | −0.030 | $5.1 \times 10^{-11}$ | −0.065 | 0.076 | 0.018 | $4.7 \times 10^{-9}$ |
| rs143840904* | chr11:2,792,092 | T/C | 1.78 | intron | KCNQ1 | $1.3 \times 10^{-5}$ | −0.11 | 0.042 | 0.057 | $2.0 \times 10^{-17}$ | −0.26 | $1.6 \times 10^{-14}$ |
| rs41286560 | chr14:100,883,117 | T/G | 3.20 | msns | RTL1 | 0.078 | −0.031 | $2.2 \times 10^{-8}$ | −0.12 | 0.0017 | 0.067 | $7.4 \times 10^{-10}$ |

Minor/Major: Minor allele and major allele. MAF (%): Minor allele frequency in percentages. Impact: divided into fs: frameshift, msns: missense, intron:intronic or intergen:intergenic. Paternal: association analysis for paternally inherited alleles. Maternal: association analysis for maternally inherited alleles. Pat. versus Mat.: difference between the effects of the paternally and maternally inherited allele. P: P value. β (s.d.): effect in standard deviation units (1 s.d.$_{male}$ = 6.9 cm, 1 s.d.$_{female}$ = 6.3 cm). For each variant, the significance threshold for concluding that a parental allele has an effect in a direction opposite to the genome-wide significant parental allele, was set at $P < 0.05/4$.
*Previously reported in a Sardinian population[29]. A chi-squared test was used to calculate P-values.

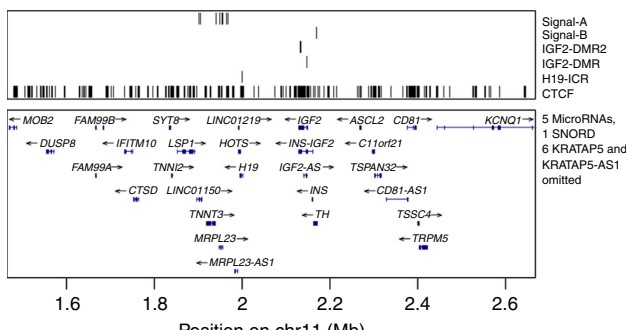

**Figure 1 | Schematic diagram showing the position (hg38) of variants corresponding to height association signals (Signal-A and Signal-B).** The diagram depicts IGF2-H19 imprinted domain in relation to the imprinting control region of the domain (H19-ICR), imprinted differentially methylated regions (IGF-DMR and IGF-DMR2) and predicted CTCF binding sites. Signal-A corresponds to rs147239461, rs113666865, rs79515490, chr11:1940471, rs76592672, rs75711836, rs145861779, rs141703487, rs75676658 and Signal-B to the variant rs7482510 (Supplementary Data 6 for detail).

Supplementary Table 8), indicating that rs143840904 is the causative variant. rs143840904 is located in a EZH2 transcription factor binding site (Supplementary Data 6). Interestingly, EZH2 is the functional enzymatic component of polycomb repressive complex 2 (PRC2), which along with the paternally expressed long noncoding RNA KCNQ1OT1 participates in maintaining monoallelic expression at the KCNQ1 imprinted domain[40,41].

The final parent-of-origin variant is a low frequency missense variant (rs41286560[T], MAF = 3.20%, p.Pro558Thr) in RTL1 on chromosome 14q32, leading to an amino acid substitution of proline (non-polar) to threonine (polar). This variant is associated with less height when paternally inherited ($\beta_{pat} = -0.12$ s.d. (−0.8 cm), $P_{pat} = 2.2 \times 10^{-8}$) (Table 2, Fig. 2 and Supplementary Fig. 16). Interestingly, rs41286560[T] has the opposite effect and increases height when maternally inherited ($\beta_{mat} = 0.067$ (0.4 cm), $P_{mat} = 0.0017$, $P_{pat \, versus \, mat} = 7.4 \times 10^{-10}$) (Table 2). rs41286560 has no strong correlates in our data or in the 1,000 Genomes phase 1 data (all $r^2 < 0.57$; SNP SNAP, https://www.broadinstitute.org/mpg/snap/ldsearch.php; accessed 09.02.2016) and is therefore likely to be the causative variant. RTL1 is known to be a paternally expressed gene located within the DLK1-MEG3 imprinted gene cluster (GeneImprint, http://www.geneimprint.com; accessed 10.10.2015) that corresponds to the ovine Callipyge locus[42]. Paternal Rtl1 knockout mice suffer from growth retardation[43].

Association of sequence variants with opposing parent-of-origin effects can be missed under the additive model. Indeed, in our data these four parent-of-origin association signals' variants would not have been detected under the additive model ($P_{add} > 1.2 \times 10^{-5}$) (Table 2). This exemplifies the value of being able to test for parent-of-origin effect and shows that the ability to test separately the chromosomes of the two parents is sometimes needed to assess effects accurately.

**Determining the cause of parent-of-origin effects.** Parent-of-origin effects could be caused by genomic imprinting of the transmitted alleles or by external effects attributable to the full genotypes of the parents – that is both the transmitted and non-transmitted alleles[44–47]. An obvious example of the latter is the influence of maternal uterine environment on birth length[48]. To disentangle these two possible causes of parent-of-origin effects, we assessed the relative impact of the maternal and paternal transmitted and non-transmitted alleles on adult height and birth length for the four parent-of-origin association signals (Supplementary Table 9). This analysis was limited to individuals who had directly genotyped parents and height measurements and/or birth length measurements. The effects of the transmitted allele remained in this reduced dataset, but we did not observe an effect of the non-transmitted alleles (Supplementary Table 9). This is consistent with parent-of-origin effects resulting from genomic imprinting rather than the full genotypes of the parents.

To assess the effect of the height associated variants identified under parent-of-origin models on gene expression, we scrutinized data from the Genotype-Tissue Expression (GTEx) project, available for multiple tissues. In GTEx (ref. 49) (analysis release V6, http://www.gtexportal.org; accessed 14.07.2016), none of the four variants rs143840904, rs147239461, rs41286560 or rs7482510 had a significant eQTL with a neighbouring gene (cis window defined as ± 1 Mb around gene transcript start site, FDR < 0.05).

In addition, a cis-eQTL analysis was performed in our data based on RNA sequencing of blood ($N = 1,990$) and adipose tissue ($N = 675$) samples. We assessed 125 genes within ± 500 Kb of the variants corresponding to the four parent-of-origin signals. Association was tested under four different models, additive, paternal, maternal and paternal versus maternal. At a significance threshold of $1.0 \times 10^{-4}$ ($P < 0.05/4 \times 125$) we did not observe significant associations of the parent-of-origin variants with gene expression in blood or adipose tissue.

**Birth length.** We observed a positive association between birth length and adult height in Iceland ($R^2 = 0.11$, $N = 539$) (Supplementary Fig. 17), consistent with previous results[50]. To examine to what degree adult height variants collectively explain birth length, we computed adult height polygenic scores for chip-typed Icelanders based on adjusted GWAS scores from the GIANT (ref. 5) study (excluding Icelandic data) (Methods). In our data these polygenic scores explain 16.4% of the variance in adult height ($P < 1.0 \times 10^{-300}$, $N = 80,546$) and 2.0% of birth

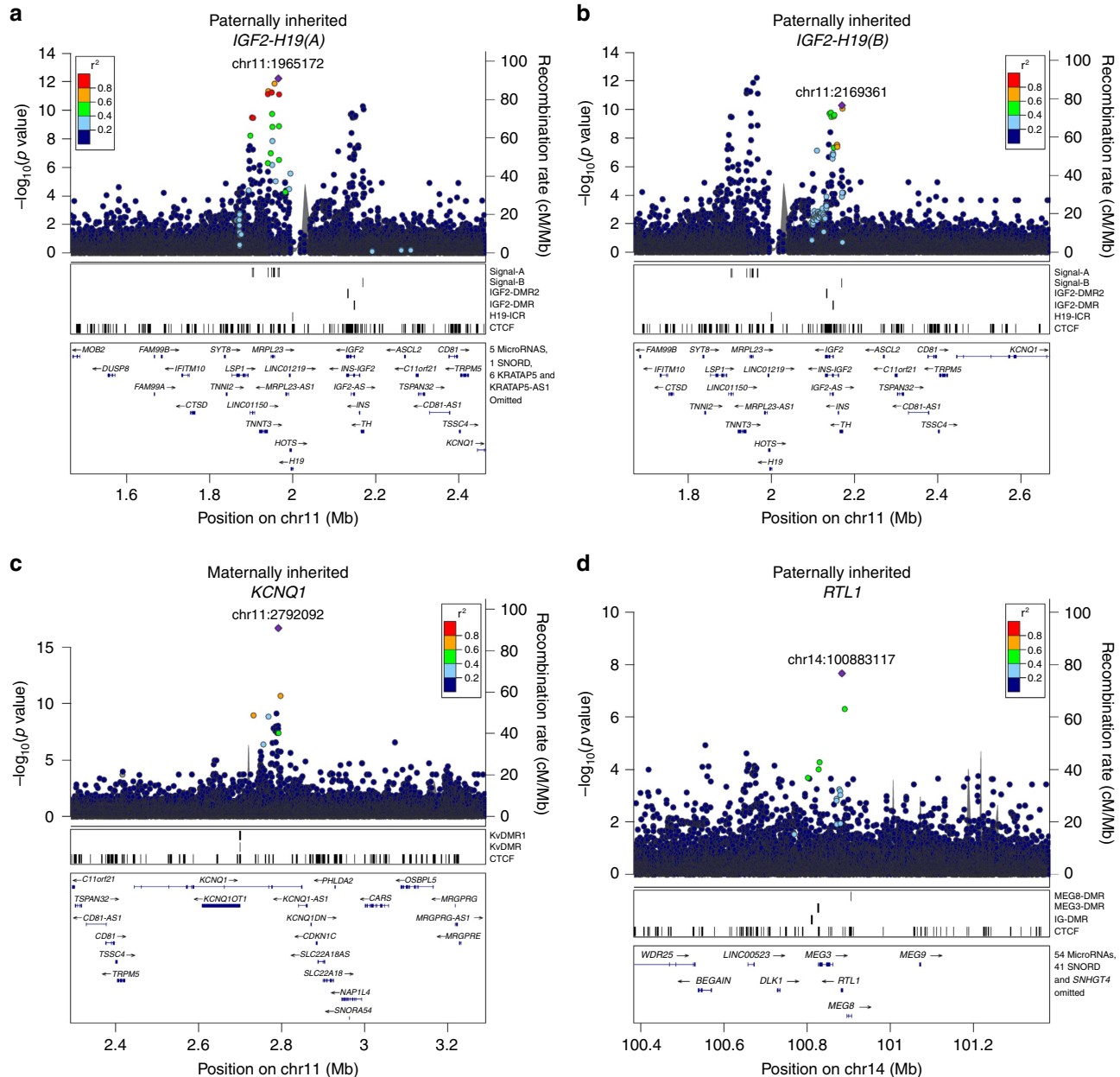

**Figure 2 | Locus-plots corresponding to four parent-of-origin signals associated with adult height.** (**a**) IGF2-H19(A), (**b**) IGF2-H19(B), (**c**) KCNQ1 and (**d**) RTL1. The plots depict association results (*P*-values) with adult height, based on a dataset of 88,835 Icelanders with height information. For each plot, the *y*-axis shows the $-\log_{10}$ *P*-values and *x*-axis the genomic position (hg38). The leading variants of independent signals are labelled as diamonds and coloured purple. Other variants are coloured according to correlation ($r^2$) with the leading marker. A chi-squared test was used to calculate *P*-values.

length ($P = 2.7 \times 10^{-20}$, N = 4,275; a *t*-test was used to calculate *P*-values.) (Supplementary Fig. 18).

The 63 variants found to be associated with adult height in this study were tested for association with birth length under the appropriate models (Supplementary Data 2). Two variants, rs147239461[T] at *IGF2-H19(A)* and rs558226101[T] in *TET1*, were found to be significant using a threshold of $7.9 \times 10^{-4}$ ($P < 0.05/63$). As in the case of adult height, rs147239461[T] reduces birth length when paternally inherited ($P_{pat} = 6.6 \times 10^{-4}$, $\beta_{pat} = -0.25$ s.d. ($-0.6$ cm)) and increases birth length when maternally inherited ($P_{mat} = 4.1 \times 10^{-2}$, $\beta_{mat} = 0.14$ s.d. (0.4 cm), $P_{pat\,versus\,mat} = 1.0 \times 10^{-4}$). In s.d. units, the effect of rs147239461 on birth length is twice its effect on adult height (Supplementary Data 2, Fig. 3). This is consistent with

observations that the imprinted *IGF2-H19* gene cluster on chromosome 11p15 is central for the control of fetal and postnatal growth[20]. The second variant, rs558226101[T], a rare missense variant in the *TET1* gene, associates with both adult height ($\beta_{adult} = 0.48$ s.d. (3.2 cm), $P_{adult} = 1.6 \times 10^{-8}$, Table 1, Supplementary Data 5 and Supplementary Fig. 3) and birth length ($\beta_{birth} = 0.86$ s.d. (2.2 cm), $P_{birth} = 5.3 \times 10^{-4}$, Supplementary Data 2) under the additive model. This rare variant causes a missense change in the *TET1* gene (MAF = 0.13%, p.Arg1783Trp), has no strong correlates in our data (all $r^2 < 0.79$) and is not present in the 1000 Genomes phase 1 data (SNP SNAP, https://www.broadinstitute.org/mpg/snap/ldsearch.php; accessed 09.02.2016). *TET1* has not been reported to affect height in humans (GWAS catalogue v1.0,

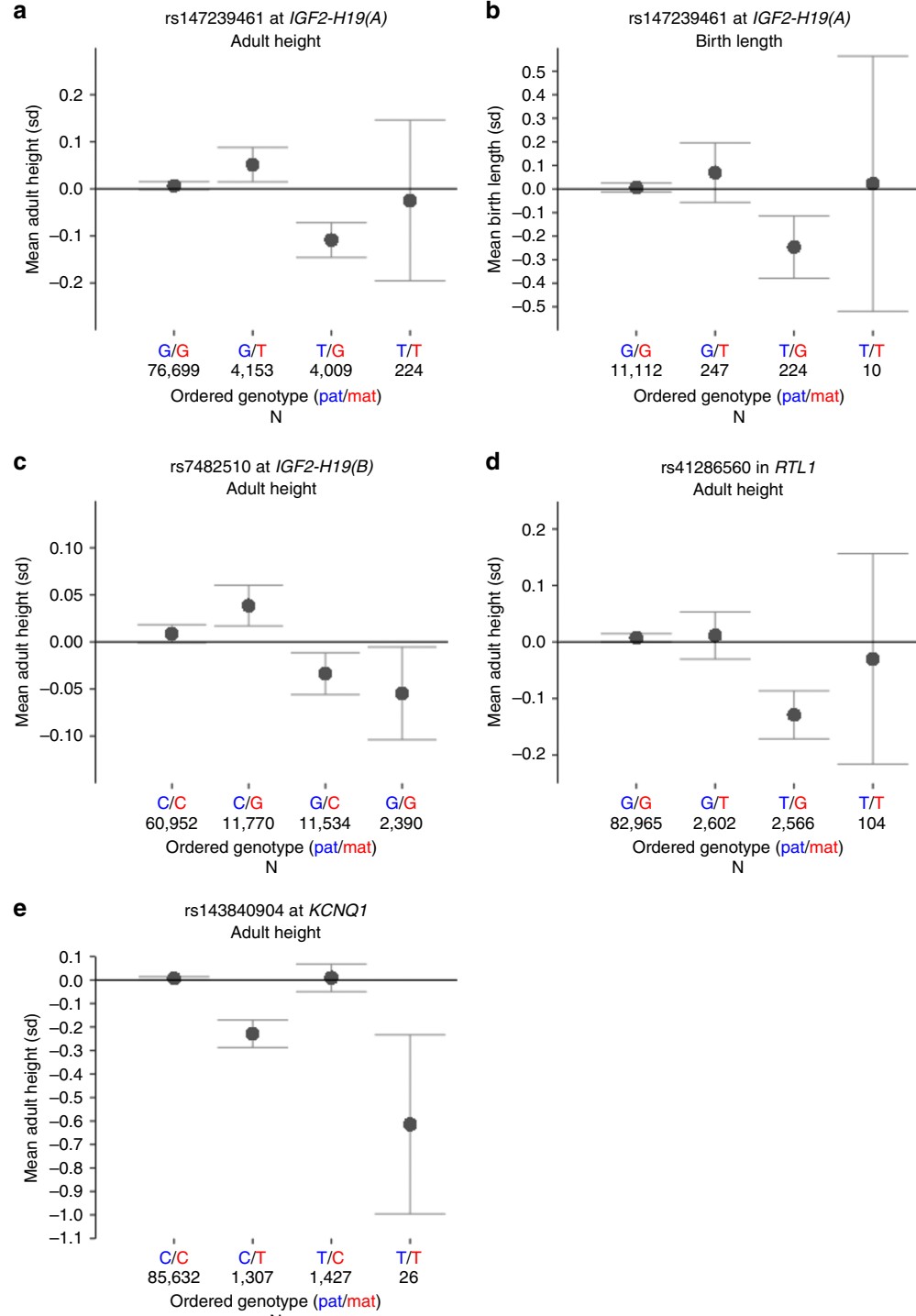

**Figure 3 | Ordered genotype height effects of the four variants identified in the present study under a parent-of-origin model.** The x-axis in each plot shows the four possible ordered genotypes with the first allele being inherited from the father and the second from the mother. The y-axis shows adult height (s.d.) in plots (**a**,**c**–**e**) and birth length (s.d.) in plot (**b**). Means were computed from the groups of imputed and family imputed Icelanders for which we have adult height and/or birth length measurements. Ordered genotypes were assigned to individuals based on allele probabilities (see Methods). Each grey dot represents the mean for individuals with the ordered genotype in question and the error-bars represent 95% confidence interval of the mean, taking the correction factor into account (Methods). The number of individuals behind the computations of each mean is shown on x-axis below each ordered genotype. Axes are not the same scale.

https://www.ebi.ac.uk/gwas/docs/downloads; accessed 04.08.2015) but the association of this missense variant with adult height and birth length reported here supports a role of *TET1* in growth regulation in humans, beginning during *in-utero* development and probably continuing after birth. *TET1* is reported to play a critical role in genomic erasure of paternal imprints in the female germ line[51]. Furthermore, *Tet1* paternal knockout mice suffer placental, fetal and postnatal growth defects as a result of hypermethylation of differentially methylated regions of paternally imprinted genes[51].

## Discussion

Among 13 novel height associations, the association is primarily with the paternally inherited allele (two at *IGF2-H19* and one in *RTL1*) in three instances, and one of the variants associates with height under the recessive model (*LECT2*). The other nine novel signals associate with height under an additive model and eight of them are represented by low frequency or rare sequence variants.

By examining the effect of non-transmitted alleles, we conclude that the effects of the parent-of-origin variants described here are mediated through imprinting. This conclusion is further supported by the fact that all four variants are located within known imprinted regions[17] (GeneImprint, http://www.geneimprint.com; accessed 10.10.2015). Although imprinted genes comprise a small subset of all genes ($\sim$1%, $N = 284$) (Supplementary Data 1), they have been reported to have a pivotal role in growth and development[20,23]. Three of the four parent-of-origin association signals fall within the well-studied 11p15 imprinted region that comprises two neighbouring imprinted domains, *IGF2-H19* and *KCNQ1*, which contain both maternally and paternally expressed genes. The paternally expressed gene *IGF2* is one of the best-characterized imprinted genes that plays a key role in the regulation of cell proliferation and growth[20,52].

Generally, it is assumed that parent-of-origin imprinting leads to either paternal or maternal mono-allelic gene expression[53]. Alternatively, there could be incomplete gene expression, with genes from both parental chromosomes expressed but to different degrees[54,55]. In addition to these patterns based on complete or partial silencing of gene expression, more complex patterns of effects have been observed[23,26]. Interestingly, for two of the variants we report, rs147239461[T] at *IGF2-H19(A)* and rs41286560[T] in *RTL1* at *DLK1*-MEG3, the minor allele is associated with less adult height when paternally inherited and greater height when maternally inherited. The same kind of effect for rs147239461[T] at *IGF2-H19(A)* was also observed with birth length indicating that this phenomenon is already at play *in utero*. Furthermore, we previously reported a distinct 11p15 variant associated with type 2 diabetes, where the minor allele confers risk when paternally inherited but is protective when maternally inherited[14]. Differential silencing of genes within the *IGF2-H19* domain is normally tightly regulated and under the control of a differentially methylated ICR making it vulnerable to genetic and epigenetic variations. rs147239461 at *IGF2-H19(A)* is located downstream from *H19* 35 Kb from the H19-ICR. Therefore, it is unlikely that the variant directly disrupts methylation of the ICR. However, since the variant and three of its strong correlates are located at a CTCF binding site it might affect methylation at the ICR and thus disturb the balanced expression of imprinted genes in the region.

*RTL1* (Retrotransposon-like 1) on chromosome 14p32 is an intronless gene with an overlapping maternally expressed antisense transcript, which contains seven miRNAs regulating the expression of this gene through an RNA interference mediated mechanism[43]. We note that the height-associated missense variant (rs41286560) with parent-of-origin effect is located 42 bp from *miR127* that has been linked to placentomegaly in mice when maternally knocked out[43]. Given that the effect is predominantly from the paternal chromosome we find it more likely that the effect is mediated through missense in *RTL1* itself. Paternal *Rtl1* knockout mice develop growth retardation at the fetal stage that persists into adulthood[43], but to our knowledge *RTL1* has not been linked to growth defects in humans.

In animal models, the influence of parent-of-origin effects has been observed in traits associated with growth or allocation of nutritional resources, and a large portion of imprinted genes are reported to have effects on the placenta[32–34]. This has been taken to support the evolutionary hypothesis of a conflict between the maternal and paternal genomes in offspring prior to weaning, whereby fathers use imprinting to increase nutritional demands at the expense of the mother, who uses imprinting to avoid excessive demands on resources[56–58]. In line with this hypothesis, it has been argued that reciprocal imprinting of the growth enhancer gene *IGF2* and the neighbouring growth suppressor gene *H19* at 11p15 represents a mechanism that is consistent with the parental conflict theory and has been widely used as the basis for its interpretation. Interestingly, imprinting disorders involved in both retarded and excessive growth map to chromosome 11p15 and are linked to depressed versus elevated expression of *IGF2* (ref. 20). In knockout mice, paternal *Igf2* knockouts are growth retarded, whereas embryos overexpressing *Igf2* exhibit overgrowth[59,60]. Taken at face value, our results seem contrary to the expectations of parental conflict through imprinting. Rather than paternal alleles increasing growth and maternal alleles decreasing growth, the paternally inherited minor alleles at *IGF2* and *RTL1* result in reduced adult height, and in the case of rs147239461 at *IGF2-H19* reduced birth length.

Taken altogether our results demonstrate that common variations affect human growth by parental imprinting in humans. Moreover, by studying adult height variants and birth length, we demonstrate that genomic imprinting is not restricted to effects *in utero*. Additionally, the unusual imprinting patterns observed in this study raise questions of what mechanism of imprinting at the molecular level can create opposite effects depending on the parent-of-origin.

## Methods

**Population.** Height measurements from 88,835 Icelanders were collected in deCODE's obesity, cancer and resident assessment instrument studies. Height was either measured using a stadiometer with the subjects wearing no shoes or self-reported on questionnaires by individuals. Birth length measurements were collected from 12,645 Icelanders in deCODE's obesity study. This group has very little overlap with the group of individuals with adult height measurements ($N = 539$).

These studies were approved by the Data Protection Commission of Iceland and the National Bioethics Committee of Iceland. Written informed consent was obtained from all participants. Personal identifiers associated with phenotypic information and samples were encrypted using a third-party encryption system.

**Genotyping and imputation.** Genotyping and imputation methods and the association analysis method in the Icelandic samples are described[28], with some modifications as follows:

*Sequencing, genotype calling and annotation.* The whole genomes of 8,453 Icelanders were sequenced using Illumina technology to a mean depth of at least 10X (median 32X). SNPs and indels were identified and their genotypes were called for all samples simultaneously using the Genome Analysis Toolkit *HaplotypeCaller* (GATK version 3.3.0)[61]. Genotype calls were improved by using information about haplotype sharing, taking advantage of the fact that all the sequenced individuals had also been chip-typed and long range phased. The sequence variants identified in the 8,453 sequenced Icelanders were then imputed into 150,656 Icelanders who had been genotyped with various Illumina SNP chips and their genotypes phased using long-range phasing[14,62].

*Imputation of variants.* Imputation of untyped variants into the mix of typed variants is now regular procedure in human genetics[63]. These imputations are usually based on local linkage disequilibrium (LD) and work well for common variants, but they are not reliable for low-frequency variants and rarely work for rare variants[6]. The long-range phasing of 150,656 Icelanders genotyped for 654,788 autosomal SNPs using Illumina chips increases imputation accuracy and speed by removing uncertainty of phasing. Of variants with a MAF over 0.1%, 96.7% were imputed with information over 0.8.

*Genealogy and imputation.* Using genealogic information, the sequence variants were imputed into un-typed relatives of the chip-typed to further increase the sample size for association analysis and increased the power to detect associations. Individuals with height measurements were either chip-typed individuals ($N = 80,546$) or first and second degree relatives of chip-typed individuals that were not chip-typed themselves ($N = 8,289$). The group of individuals with birth length measurements consisted of chip-typed individuals ($N = 4,275$) and first and second degree relatives of chip-typed ($N = 8,370$). A total of 31.6 million variants were used in the association analysis under an additive model and parent-of-origin

models. Both parent-of-origin models (paternal and maternal) were tested for variants for which Icelandic genealogy was used to assign parental origin to phased haplotypes Long-range phasing of haplotypes using surrogate parents allows for accurate phasing of Icelandic samples and the Icelandic genealogy coupled with the large fraction of chip-typed individuals in the population enabled the determination of the parent-of-origin for the genotypes (refs 6,14). Initially this method was applied to a set of 38,167 Icelanders (ref. 6), in the current study we apply the same methodology on a larger set of chip-typed individuals ($N = 150,656$). For parent-of-origin models, we note that the number of variants that reside in or are adjacent to ($\pm 250$ kb) known or predicted imprinted genes ($N = 284$, corresponding to 1% of the genome) was 1.3 million (4.1% of all variants tested). For recessive analysis, the number of tested variants for which we had homozygotes for the minor allele was 19.2 million. All of the tested variants had imputation information over 0.8.

To enrich for very rare variants affecting height, we included in our sequencing all individuals in the dataset who deviated more than three standard deviations (s.d.) from the mean value of adult height in the sample set ($N = 165$) (Supplementary Fig. 1).

**Sample preparation and DNA whole-genome sequencing methods.** Our dataset contains samples obtained using three different library preparation methods from Illumina. In addition sequencing was performed using three different types of Illumina sequencing instruments.

(a) Standard TruSeq DNA library preparation method. Illumina GAIIx and/or HiSeq 2,000 sequencers.
(b) TruSeq DNA PCR-free library preparation method. Illumina HiSeq 2,500 sequencers.
(c) TruSeq Nano DNA library preparation method. Illumina HiSeq X sequencers.

Sample preparation and sequencing using the standard TruSeq DNA library preparation method. Approximately 1 µg of genomic DNA, isolated from frozen blood samples, was fragmented to a mean target size of ~300–400 bp using a Covaris E210 instrument. The resulting fragmented DNA was end repaired using T4 and Klenow polymerases and T4 polynucleotide kinase with 10 mM dNTP followed by addition of an 'A' base at the ends using Klenow exo fragment (3′ to 5′-exo minus) and dATP (1 mM). Sequencing adaptors containing 'T' overhangs were ligated to the DNA products followed by agarose (2%) gel electrophoresis. Fragments of about 450–500 bp were isolated from the gels (QIAGEN Gel Extraction Kit), and the adaptor-modified DNA fragments were PCR enriched for ten cycles using Phusion DNA polymerase (Finnzymes Oy) and a PCR primer cocktail needed for paired-end sequencing. Enriched libraries were purified using AMPure XP beads. The quality and concentration of the libraries were assessed with the Agilent 2,100 Bioanalyzer using the DNA 1,000 LabChip. Libraries were stored at −20 °C. Sequencing-by-synthesis (SBS) was performed on either Illumina GAIIx or HiSeq 2,000 instruments, respectively. Paired-end libraries were sequenced by $2 \times 76$, $2 \times 101$ or $2 \times 120$ cycles of incorporation and imaging with Illumina SBS kits, TruSeq v5 for the GAIIx. For the HiSeq 2,000, $2 \times 101$ cycles with SBS kits v2.5 or v3 were employed. Each library was initially run on a single lane on a GAIIx for validation, assessing optimal cluster densities, insert size, duplication rates and comparison to chip genotyping data. Following validation, the desired sequencing depth (10X to 30X) was then obtained using either sequencing platform. Targeted raw cluster densities ranged from 500–800 K⁻¹ mm⁻², depending on the version of both the sequencing chemistry and the data imaging/analysis software packages (SCS2.8/RTA1.8 or SCS2.9/RTA1.9 for the GAIIx and HCS1.3.8. or HCS1.4.8 for HiSeq 2,000). Real-time analysis involved conversion of image data to base-calling in real-time.

Sample preparation and sequencing using the TruSeq DNA PCR-free method. Paired-end libraries for sequencing were prepared according to the manufacturer's instructions (Illumina, TruSeq DNA PCR-free). In short, ~1 µg of genomic DNA, isolated from frozen blood samples, was fragmented to a mean target size of 350 bp using a Covaris E210 ultrasonicator followed by clean-up using AmPure XP purification beads. Blunt-end DNA from the resulting fragments was generated using a mix of 3′>5′ exonuclease and 5′>3′ polymerase activities, respectively, followed by 5′-phosphorylation using T4 polynucleotide kinase. Size-selection of the blunt-end fragments was done using a two-step purification strategy with different ratios of the AmPure XP purification beads (0.6X and 1X). Finally, 3′-adenylation and ligation of barcoded adaptors was performed, followed by clean-up with magnetic beads. The quality and concentration of the libraries were assessed with the Agilent 2,100 Bioanalyzer using the DNA 1,000 LabChip (Agilent). Barcoded libraries were stored at −20 °C. All steps in the workflow were monitored using an in-house laboratory information management system with barcode tracking of all samples and reagents. All samples were first pooled (12–24 plex) and sequenced on Illumina's MiSeq instruments ($2 \times 25$ cycles) to assess quality and effective concentration of sequencing libraries. Subsequent deep sequencing was done on HiSeq 2,500 instruments, were each sample was sequenced on 3 lanes, generating >100 Gb of raw data and at least 30X coverage. Sequencing was done using TruSeq v3 reagents, paired-end $2 \times 100$ cycles. System operation and base calling in real-time was done using HCS 2.2.38 and RTA 1.18.61.

Sample preparation and sequencing using the TruSeq Nano DNA method. The sample preparation workflow was essentially the same as described above for the TruSeq DNA PCR-free method, except the input amount was 100 ng of genomic DNA (instead of 1 µg) and following clean-up of adaptor ligated DNA, the samples were enriched by 8-cycles of PCR using a PCR primer cocktail, followed by Ampure XP bead clean-up. The quality and concentration of the libraries were assessed with the Perkin Elmer LabChip GX instrument using the HT DNA HiSens reagent kit. Sequencing was done using the HiSeq X HD reagent kit. Each sample was loaded onto the HiSeq X instrument at a concentration of 300 pM and sequenced to high depth (>30X). System operation and base calling in real-time was done using HCSX 3.1.26 and RTA2 2.3.9.

**Association analysis.** Adult height measurements and birth length measurements were corrected for year of birth and standardized separately for each of the sexes to have a standard normal distribution. Measured and self-reported heights were corrected separately.

*Genetic models.* Four different genetic models were tested: additive, recessive and parent-of-origin (paternal and maternal). Parent-of-origin models were performed by testing the paternal and maternal alleles separately. For SNPs that associated with height under multiple models, the appropriate model was concluded to be the one that gave the most significant height association. All tests reported in the present study are two-sided.

*Quantitative trait association testing.* A generalized form of linear regression was used to test for association of adult height and birth length with SNPs. Let $y$ be the vector of quantitative measurements, and let $g$ be the vector of expected allele counts for the SNP being tested. We assume the quantitative measurements follow a normal distribution with a mean that depends linearly on the expected allele at the SNP and a variance covariance matrix proportional to the kinship matrix:

$$\mathbf{y} \sim \mathcal{N}\left(\alpha + \beta\mathbf{g}, 2\sigma^2\mathbf{\Phi}\right), \tag{1}$$

Where

$$\Phi_{ij} = \begin{cases} \frac{1}{2}, & i=j \\ 2k_{ij}, & i \neq j \end{cases}. \tag{2}$$

is the kinship matrix as estimated from the Icelandic genealogical database. It is not computationally feasible to use this full model and we therefore split the individuals into smaller clusters. The maximum likelihood estimates for the parameters $\alpha$, $\beta$ and $\sigma^2$ involve inverting the kinship matrix. If there are $n$ individuals in the cluster, then this inversion requires $O(n^3)$ calculations, but since these calculations only need to be performed once the computational cost of doing a genomewide association scan will only be $O(n^2)$ calculations; the cost of calculating the maximum likelihood estimates if the kinship matrix has already been inverted.

*LD score regression.* To account for inflation in test statistics due to cryptic relatedness and stratification, we applied the method of LD score regression[64]. With a set of 1.1 M variants we regressed the $\chi^2$ statistics from our GWAS scan against LD score and used the intercept as a correction factor. The LD scores were downloaded from a LD score database (ftp://atguftp.mgh.harvard.edu/brendan/1k_eur_r2_hm3snps_se_weights.RDS; accessed 23.06.2015). The estimated correction factor for adult height was 1.48 for the additive model, 1.15 for the recessive model, 1.22 for the maternal model and 1.23 for the paternal model (Supplementary Fig. 19). The estimated correction factor for birth length was 1.04 for the additive model, 1.01 for the recessive model, 1.02 for the maternal model and 1.03 for the paternal model.

*Significance thresholds.* The threshold for genome-wide significance was corrected for multiple testing using a class-specific Bonferroni procedure based on predicted functional impact of classes of variants[35] (Supplementary Fig. 2). This yielded significance thresholds of $1.7 \times 10^{-6}$ for high-impact variants (including stop-gained, frameshift, splice acceptor or donor, $N = 9,989$), $9.8 \times 10^{-8}$ for moderate-impact variants (including missense, splice-region variants and in-frame indels, $N = 170,692$) and $5.3 \times 10^{-10}$ for low-impact variants ($N = 31,421,778$). For the parent-of-origin association analysis, to test if an parental allele had effect in a direction opposite to the genome-wide significant parental allele, the threshold of significance was corrected for the number of genome-wide significant signals observed in our data under the parent-of-origin models (threshold = 0.05/4).

*Independent signals.* We performed conditional analysis for each $\pm 2$ Mb area that contains at least one variant with genome-wide significant association with height in our data. All variants in the area with info > 0.9 were included in the analysis with the lead variant (lowest $P$-value) as covariate. Variants were concluded to belong to an independent signal if their adjusted $P$-value was genome-wide significant. Conditional analysis was repeated for each area until a result with no genome-wide significant adjusted $P$-value was attained.

*Classification.* To distinguish between reported, refinement and novel signals, we conducted conditional analysis for each variant that represented a genome-wide significant association in our data and reported height associated variants within $\pm 500$ Kb. The group of reported variants consisted of (i) the 697 SNPs reported by GIANT (ref. 5) to associate with height under the additive model, (ii) one SNP reported by Steinthorsdottir et al.[65] to associate with height under the additive model, (iii) six SNPs reported by Zoledziewska et al.[29] to associate with height when maternally inherited and (iv) all variants in entries containing the word 'height' in the column: DISEASE/TRAIT in the GWAS catalogue (GWAS catalogue

v1.0, https://www.ebi.ac.uk/gwas/docs/downloads; accessed 04.08.2015). Variants were classified as reported if published variants could account for their effect. Variants were classified as a refinement of a formerly reported signal if they fulfilled all the following three criteria: (i) the refinement variant is moderately correlated with at least one reported variants $(0.10 < r^2 < 0.80)$, (ii) the refinement variant demonstrates a significantly stronger association and accounts for the effects of its reported correlates (based on conditional analysis) and (iii) the reported correlates could not fully account for the effect of the refinement variant detected in the present study (adjusted $P < 1.0 \times 10^{-3}$) (based on conditional analysis). We concluded a variant to be novel if it was not correlated ($r^2 < 0.10$) to any variant reported to associate with height under the same model (additive, recessive, maternal, paternal) and conditional analysis revealed that reported height-variants could not account for its effect.

*Fraction of variance explained.* For computation of fraction of variance explained, we chose only one SNP representing each independent signal. For each variant associating with height under the additive model, the fraction of variance explained was calculated using the formula $2f(1-f)a^2$, in which $f$ denotes the minor allele frequency of the variant and $a$ is the additive effect. For variants associating under the recessive model the formula $f_h(1-f_h)a_r^2$ was used, in which $f_h$ denotes the homozygous frequency of the variant and $a_r$ denotes the recessive effect. For variants associating under parent-of-origin models, fraction of variance explained was computed using the formula $f(1-f)(a_m^2 + a_p^2)$ where $f$ denotes the minor allele frequency of the variant, $a_m$ denotes the effect under the maternal model and $a_p$ denotes the effect under the paternal model. Both parent-of-origin effects, $a_m$ and $a_p$, were only included in the formula if the variant associates with height under both models with a $P < 1.0 \times 10^{-2}$. If that is not the case, then the genome-wide significant parent-of-origin effect was only included in the formula and the other parent-of-origin effect was given the value of zero.

*Testing nontransmitted alleles.* To determine the cause of parent-of-origin effects, non-transmitted alleles were tested with a simple linear regression for each SNP associating with height under a parent-of-origin model. This analysis was limited to chip-typed individuals who had chip-typed parents and height measurements and/or birth length measurements.

*Polygenic scores.* GWAS results for 2,550,858 variants were computed based on GIANT (ref. 5) data after having removed Icelandic samples. Imputed genetic variants were available for 80,546 Icelanders with adult height measurements (adjusted for sex and age) and 4,275 Icelanders with birth length measurements (adjusted for sex and age). Of the GWAS variants which were also available in the imputation set, 1,746,527 were biallelic SNPs and met quality control standards. LDpred[66] was used to adjust for the correlation among the effects of these variants due to linkage disequilibrium and these adjusted effects were used to produce the adult height polygenic scores. Simple linear regressions were then performed to determine the association of adult height and birth length with the polygenic scores.

*Genotypic effect.* Ordered genotypes for four variants were assigned to imputed and family imputed individuals with adult height measurements and/or birth length measurements. Imputation information had to be more than 0.8 for both alleles for the variant in question in order for an individual to be included in the analysis. Mean adult height and/or mean birth length were computed for the four ordered genotypes and the results are displayed in Fig. 3. 95% confidence intervals for the means were computed with the formula: mean $\pm 1.96 \times$ se $\times \sqrt{\text{cf}}$ in which s.e. is the standard error of the mean and cf is the correction factor for the additive model, 1.48 for adult height and 1.04 for birth length.

*Contig at IGF2-H19 locus.* We observe a cluster of missing variants at the IGF2-H19 locus spanning 5 kb. This is due to a non-uniquely aligning contig that is assigned a mapping quality of zero during variant calling and thus excluded from subsequent analysis.

*Muscle-specific promoters.* Muscle-specific promoters were defined by taking the union of regions annotated as a 'Tss' chromatin state in any of the eight muscle reference epigenomes from the Roadmap Epigenomics Project[67] and subtracting regions that are in a 'Tss' chromatin state in any other reference epigenome.

**Data availability.** Summary level data of markers tested for association is described in Scientific data as of 2015 with the identifier http://dx.doi.org/10.1038/sdata.2015.11[68]. Whole genome sequencing summary data are available at the European Variant Archive (EVA) under the accession code PRJEB8636.

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

## Acknowledgements

We thank the individuals who participated in the study and whose contributions made this work possible. We thank our valued colleagues who contributed to data collection, sample handling and genotyping.

## Author contributions

S.B., A.O., A.H., R.P.K., P.S., U.T. and K.S. wrote the initial draft of the manuscript. S.B., A.O., A.K., M.L.F., B.O.J., O.B.D, G.S., G.T., G.A.A., L.D.W., J.K.S., P.D.I., D.F.G., H. Helgason and P.S. analysed the data. S.B., A.O., D.F.G., H. Helgason, A.H., S.A.G., A.O., A.K., K.F.A. and G.M. created methods for analysing the data. H. H, O.B.D. and S.A.G. performed the experiments. H.H., L.T., T.R., G.B.W., G.S., collected the samples and information. D.F.G, G.M, U.T., A.H., P.S. and K.S. designed the study.

## Additional information

**Competing financial interests:** The following authors affiliated with deCODE genetics/Amgen are employed by the company: S.B., A.O., A.H., R.P.K., G.S., A.O., G.T., O.B.D., G.A.A., G.S., B.O.J., H.H., K.F.A., G.B.W., S.A.G., P.D.I. L.D.W., J.K.S., M.L.F., T.R., A.K., G.M., H. Helgason, U.T., D.F.G., P.S. and K.S.

**Publisher's note**: 

