## [Peer Review File · Nature Communications]

Reviewers' Comments:

Reviewer #1 (Remarks to the Author)

Major comments:

Benonisdottir and colleagues present a large study on epigenetic and genetic components of height regulation. They find 4 novel loci whose effect act in a parent-of-origin fashion. The study is well-designed, the analysis is rigorously conducted and the results are convincing to me. There are however, several aspects that could improve the study.

In my view the novel, but non parent-of-origin (PO) signals are of little relevance here and I'd completely remove them from the paper. There is no interpretation or description of those findings whatsoever and break the flow of the paper.

There are very little technical details given (the appendix has only results). For example: There are no details about the genotyping. How did they correct for batch effects? In such a large sample many artefacts can come to play. The LD score regression intercept is quite high (1.48) for height, which shows that there is clearly unaccounted stratification issue. How do the samples/batches map on the genetic PCA map?

All 4 parent of origin signals seem to have opposite effect when inherited maternally vs paternally. This is not the most usual PO signal (although has been seen in more and more studies), it is difficult to explain mechanistically. Since these findings are coming from a single cohort a single population (except for the known Sardinian hit), it would be more reassuring to check it in other populations where duos/trios are collected (Sardinia, Generation Scotland, Framingham, Alspac, etc.). Without replication we are not sure if the 3 new hits are population specific.

Related to this, there is no attempt at mechanistical explanation of these signals apart from the fact that these appear in imprinted regions. Is there allele specific expression of the transcripts at these loci (or their coding correlates)? There are many relevant eQTL data sets to look at. Given that the authors provide no functional characterization of these new SNPs I'd expect to see much deeper bioinformatics check on these (both for regulatory and coding variants differently).

Minor comments:

Line 265 (page 12): Conclusions about no association with birth weight are not justified. First, birth weight is reported with much more error, second the sample size for BW was much smaller. Therefore I don't think that any conclusion can be made with the available data.

Line 266-269 (page 12): The expected paternal height increase and maternal height decreasing effects are not easy to check here: Depends which allele one looks at. If the baseline is the homozygous minor group, then the effects look opposite. If e.g. for rs147239461 we take the reference the TT group then the effect of mat-G is negative and pat-G is positive, but if the reference is the GG group then mat-T is positive and pat-T is negative then, one can argue both ways.

Line 425 (page 19): "had a more significant association with height in our data than their reported correlates" This can be random fluctuation; this is not a solid reason to say it is a refined signal.

Reviewer #2 (Remarks to the Author)

This is a compelling story of the identification of several new variants associated with human growth, with a particular focus on four, 3 of which are novel, that do so with parent of origin effects. The power of the Decode study and its ability to impute parental genotypes and assess the effects of non transmitted as well as transmitted alleles makes the study unique. The parent of origin effects are very likely to be due to imprinting. The results build a fascinating story around the variation in the IGF2/H19 region in particular. The added interest is that the maternal effects appear to increase growth, whilst paternal effects decrease growth, contrary to the evolutionary conflict theory that never had too much evidence.

I have only minor suggestions for the authors to consider, but it looks like a well polished manuscript:

1. It would be worth stating in a few more places what the effects are in cm as well as SDs. As height is very close to normally distributed, then a simple conversion should suffice - some of the additive and recessive effects of low frequency variants look like v large effects.
2. The conclusion that "take together our results demonstrate that genomic imprinting affects variation of anthropometric traits in humans" should be qualified because we knew this. It is that the results show multiple that more common variation , especially in the IGF2/H19 region affect human growth by parental imprinting "?

Manuscript: NCOMMS-16-09989

Epigenetic and genetic components of height regulation

by Benonisdottir et al.

Reviewer #1 (Remarks to the Author):

Major comments:

Benonisdottir and colleagues present a large study on epigenetic and genetic components of height regulation. They find 4 novel loci whose effect act in a parent-of-origin fashion. The study is well-designed, the analysis is rigorously conducted and the results are convincing to me. There are however, several aspects that could improve the study.

Comment 1.:

- a) In my view the novel, but non parent-of-origin (PO) signals are of little relevance here and I'd completely remove them from the paper.**
- b) There is no interpretation or description of those findings whatsoever and break the flow of the paper.**

Answer:

Response to sub-comment a)

Not many studies can test for parent-of-origin specific effects in GWAS. This is possible in Iceland, due to the combination of long-range phased genotypes and a comprehensive genealogical database. Due to the novelty and importance of parent-of-origin results, we chose to emphasize them in the main text of the manuscript.

However, our study design was nonetheless based on the testing of all four different models for each marker, genome-wide: additive, recessive and parent-of-origin (paternal and maternal). For each model, we identified novel signals and we prefer including all the results in the report in order to get them out to the community in a timely manner.

Response to sub-comment b)

We have added a sentence in the main text, on page 5 in paragraph 2, illustrating that the results for the additive and recessive models are discussed in more details in the Supplementary text and that the focus in the main text is on the parent-of-origin effects:

“The results for the novel additive and recessive signals are discussed in more details in the Supplementary text whereas the focus of the main text is on the parent-of-origin signals.”

With the results for the additive and the recessive models presented in this manner we do not think that it breaks up the flow of the paper.

Comment 2:

- a) **There are very little technical details given (the appendix has only results). For example: There are no details about the genotyping.**
- b) **How did they correct for batch effects?**
- c) **In such a large sample many artefacts can come to play. The LD score regression intercept is quite high (1.48) for height, which shows that there is clearly unaccounted stratification issue. How do the samples/batches map on the genetic PCA map?**

Answer:

Response to sub-comment a)

We want to point the reviewer to the subsection “Genotyping and imputation” in Methods which is a part of the main text, where some technical details about genotyping methods are described (page 14, paragraph 4 and page 15, paragraph 1-3). It is important to note that in this subsection, we also refer to a methodological deCODE paper by Gudbjartsson et al, 2015 (PMID: 25807286) that contains extensive details about the genotyping (including whole-genome sequencing and chip-typing) and imputation methods used in our study. Incorporating all of these methods into the present manuscript would lengthen it considerably and would repeat previously published material.

To make the methods more accessible for the reader we have added the following sub headers to paragraphs in the chapter Genotyping and imputation (page 14 and 15):

Sequencing, genotype calling and annotation (page 14, paragraph 4)

Genealogy and imputation (page 15, paragraph 2)

Response to sub-comment b)

Concerning the question about batch effects, to make it clear that the sequence calling for individuals is performed simultaneously, we have made changes to the following sentence in methods (changes are in underlined text):

“genotypes were called for all samples simultaneously using the Genome Analysis Toolkit HaplotypeCaller” (page 14, paragraph 4)

Also, we would like to point out that while the chip genotyping and whole genome sequencing was performed in batches, the long-range phasing and imputation steps are

performed for all chip-typed individuals simultaneously (page 14, paragraph 4 and Gudbjartsson et al, 2015, PMID: 25807286). This approach incorporates many different quality controls to overcome batch effects and provides accurate genotype-calling – in particular, it leverages long-range haplotype sharing to validate genotype calls. In our data, we observe a high imputation accuracy, where 96.7% of variants with a minor allele frequency over 0.1 % achieve an imputation information over 0.8 which is beneficial when imputing with whole-genome sequencing data. Furthermore Gudbjartsson et al, 2015, found that the concordance for 28,204 chip-typed SNPs, which were not part of the long-range phasing set, was high (98.4% of SNPs with DAF >1% were imputed accurately, $r^2 > 0.9$) (PMID: 25807286).

To pinpoint the imputation accuracy of this approach we have added the following text in methods (page 15, paragraph 1):

“Imputation of variants. Imputation of untyped variants into the mix of typed variants is now regular procedure in human genetics (PMID: 17572673). These imputations are usually based on local linkage disequilibrium (LD) and give good results for common variants, but they are not dependable for low-frequency variants and rarely work for rare variants (PMID: 25807286). The long-range phasing of 150,656 Icelanders genotyped for 654,788 autosomal SNPs using Illumina chips increases imputation accuracy and speed by removing uncertainty of phasing. Of variants with a MAF over 0.1%, 96.7% were imputed with information over 0.8.”

Response to sub-comment c)

The high regression intercept (1.48) is due in particular to the fact that the study is based on a large set of individuals who represent a large fraction of a founder population (88,835 individuals out of 320,000 Icelanders). This will inevitably result in relatedness of individuals within the dataset. Our method of testing for association takes the closest relatedness into account using a mixed effect model, but is not able to fully account for the relatedness between individuals. The correction factor is used as a method to account for this.

We use LD score regression to reduce the effect of causative markers on the correction factor. However, again this method is not perfect and in particular will not adjust sufficiently for rare causative markers. Hence, some of the correction factor will be because of residual causal signals.

The first principal components trace the geography at Iceland. However they account for a very small proportion of phenotypic variance and adjusting for them has little impact on the correction factor.

Comment 3:

- a) All 4 parent of origin signals seem to have opposite effect when inherited maternally vs paternally. This is not the most usual PO signal (although has been seen in more and more studies), it is difficult to explain mechanistically.
- b) Since these findings are coming from a single cohort a single population (except for the known Sardinian hit), it would be more reassuring to check it in other populations where duos/trios are collected (Sardinia, Generation Scotland, Framingham, Alspac, etc.). Without replication we are not sure if the 3 new hits are population specific

Answer:

Response to sub-comment a)

Only two of the four signals that we detected under a parent of origin model have statistically significant effects in opposite directions depending on the sex of the parent, using a significance threshold of $P=0.05/4$ (see Table 2, page 36). We have added the following information about the threshold in footnotes in Table 2, page 36.

“For each variant, the significance threshold for concluding that a parental allele has an effect in a direction opposite to the genome-wide significant parental allele, was set at $P<0.05/4$.”

Not many GWAS studies are equipped to test association with maternal and paternal alleles separately. Thus, few such associations have been reported in the literature, making it difficult to evaluate what can be qualified as usual. In total, eleven associations are reported to have parent-of-origin effects (Kong et al. 2009 $N = 6$, Wallace et al. 2009, $N = 1$, Perry et al. 2014 $N = 3$, Gudbjartsson et al. 2015 $N = 1$). Among those, Kong et al. (PMID: 20016592) reported a variant with significant and opposite effects on type-2-diabetes when inherited maternally vs paternally.

Response to sub-comment b)

Concerning population specificity, we have now provided in Supplementary Table 11 the frequency in 1000 genomes and EXAC of all variants correlated with the three novel parent of origin signals (see Supplementary Table 11 in the excel supplement in the current version. It was labeled as Supplementary Table 12 in former submission).

As noted, one of the four parent-of-origin signals (rs143840904 in *KCNQ1*) was first reported in the Sardinian population. In the Icelandic population, this signal is represented by a singleton with low frequency ($MAF_{ice} = 1.8\%$). The sample size in the Sardinian population is modest ($N=6,307$) compared to the Icelandic set ($N = 88,835$) but a high frequency of the variant in Sardinians caused by a founder effect ($MAF_{Sardinia} = 9.1\%$) allowed the authors to detect this signal. The three other cohorts with parent-of-origin information available i.e. Generation Scotland (<http://www.ed.ac.uk/generation-scotland/using->

resources/resources/scottish-family-health-study), Framingham heart study (<http://bmcgenet.biomedcentral.com/articles/10.1186/1471-2156-4-S1-S76>) and ALSPAC (<http://www.bristol.ac.uk/alspac/about/>) are probably not large enough to observe significant association of the signals describe here. To replicate parent-of-origin association signals for height in other populations would require larger sample sizes of phased individuals with genealogical records to determine the parental-origin of alleles. As far as we are aware, such a sample set is not currently available outside Iceland.

We note that the variant rs7482510 at IGF2/H19(B) (MAF = 16.8%), is correlated ($r^2 = 0.49$) with a variant (rs4320932) that has been reported to be associated with height under the additive model in a large GWAS meta-analysis (GIANT), with an additive effect similar to what we observe in our data ($\text{Effect}_{\text{GIANT}} = -0.028$, $\text{Effect}_{\text{Iceland}} = -0.033$). These results demonstrate that this height association signal exists in other populations. However, their effect estimates assumes that both parental allele have the same effects, which we reject by our parent-of-origin data (paternal $\text{Effect}_{\text{Iceland}} = -0.064$ ($P = 3.0 \times 10^{-10}$), maternal $\text{Effect}_{\text{Iceland}} = 0.10$ ($P = 0.33$), Pat vs. Mat $P = 3.0 \times 10^{-7}$).

Comment 4:

- a) Related to this, there is no attempt at mechanistical explanation of these signals apart from the fact that these appear in imprinted regions.**
- b) Is there allele specific expression of the transcripts at these loci (or their coding correlates)? There are many relevant eQTL data sets to look at.**
- c) Given that the authors provide no functional characterization of these new SNPs I'd expect to see much deeper bioinformatics check on these (both for regulatory and coding variants differently).**

Answer:

Response to sub-comment a)

The fact that the parent-of-origin signals are located within imprinted regions is of central importance. In addition, we only detect signals under parent of origin models at reported imprinted loci even if the whole genome is tested under these models.

See response to sub-comment c) regarding mechanistic explanation of the parent-of-origin signals.

Response to sub-comment b)

To answer the reviewers question regarding allele specific expression, we have now added two analyses.

First, scrutinizing the Genotype-Tissue Expression (GTEx) database (<http://www.gtexportal.org>, accessed July 14, 2016) we do not observe a correlation of gene expression with the variants that associated with height under parent-of-origin models.

Second, in our data based on RNA sequencing of blood (N = 1,990) and adipose tissue (N = 675) samples, we performed a cis-eQTL analysis for 125 genes within +/- 500 Kb of the variants corresponding to the four parent-of-origin signals. We tested association of the parent-of-origin variants with gene expression in blood or adipose tissue under four different models, additive, paternal, maternal and paternal vs. maternal. At a significant threshold of 1.0×10^{-4} ($P < 0.05/4 \times 125$) we did not observe a significant association.

We now refer to these observations in the main text of the revised manuscript as follows (page 9, paragraph 2 and 3):

„To assess the effect of the height associated variants identified under parent-of-origin models on gene expression, we scrutinized data from the Genotype-Tissue Expression (GTEx) project, available for multiple tissues, In GTEx (analysis release V6, see URLs), none of the four variants rs143840904, rs147239461, rs41286560 or rs7482510 had a significant eQTL with a neighbouring gene (cis window defined as +/- 1Mb around gene transcript start site, FDR<0.05).

In addition, a cis-eQTL analysis was performed in our data based on RNA sequencing of blood (N = 1,990) and adipose tissue (N = 675) samples. We assessed 125 genes within +/- 500 Kb of the variants corresponding to the four parent-of-origin signals. Association was tested under four different models, additive, paternal, maternal and paternal vs. maternal. At a significant threshold of 1.0×10^{-4} ($P < 0.05/4 \times 125$) we did not observe significant associations of the parent-of-origin variants with gene expression in blood or adipose tissue.”

Response to sub-comment c)

We would like to point out to the reviewer that Supplementary Table 11, which we have updated in the revised manuscript, (available as excel sheet) contains regulatory information. These include, overlap with chromatin states, transcription factor binding sites and distance to imprinting control regions for the variants reported under a parent-of-origin model and their correlates. In order to increase the functional annotation, as suggested by the reviewer, we now include the cell type of the predicted chromatin states (Supplementary Table 11). Furthermore, we have revised and re-structured Supplementary Table 11 to be more complete but still readable. We now also refer appropriately to the table in the main text:

Page 6, paragraph 4; Page 7, paragraph 1 ; Page 7, paragraph 2; Page 8, paragraph 1

We note that in the main text we already attempted a mechanistic explanation for 3 of the 4 the parent-of-origin variants.

We have now added a hypothesis for the remaining one as follows:

- For rs7482510 at *IGF2/H19(B)* we added the following text on page 7, paragraph 2 (underlined text):

“The minor allele G associates with less height when paternally inherited (MAF = 16.84 %, $\beta_{\text{pat}} = -0.065$ SD, $P_{\text{pat}} = 5.1 \times 10^{-11}$) (Table 2, Fig. 2, Supplementary Fig. 4), but does not affect height when maternally inherited ($\beta_{\text{mat}} = 0.018$, $P_{\text{mat}} = 0.076$) (Table 2). The variant rs7482510 is located at an estrogen receptor (ESR1) binding site as observed in ENCODE ChIP-seq data (Supplementary Table 11). Interestingly, ESR1 is the main estrogen receptor regulating skeletal growth (PMID: 16511588). We will refer to this signal as *IGF2/H19(B)*.”

We have expanded the mechanistic hypothesis for rs147239461 at *IGF2/H19(A)* which now reads as follows:

- For rs147239461 at *IGF2/H19(A)* we added the following text on page 7, paragraph 1 (underlined text):

“Out of the nine strong correlates of rs147239461, three are located within binding sites for the highly conserved DNA-binding protein CTCF that can act as an insulator by blocking interactions between enhancers and promoters frequently involved in imprinting regulation (Supplementary Table 11). Also of note, the variant rs75676658 which is 9 Kb from the H19-ICR is located in a muscle specific promoter, a promoter type that constitutes 0.28 % of the genome (Methods), which is interesting in the light that *IGF2* has been shown to promote mesoderm formation in mammals (PMID: 11076682)”

- To clarify how we define muscle-specific promoters, we have added the following paragraph in methods (page 22, paragraph 3):

“Muscle-specific promoters were defined by taking the union of regions annotated as a “Tss” chromatin state in any of the eight muscle reference epigenomes from the Roadmap Epigenomics Project (PMID: 25693563) and subtracting regions that are in a “Tss” chromatin state in any other reference epigenome.”

- Furthermore, for rs147239461 at *IGF2/H19(A)* we wrote on page 12, paragraph 1. This has not been changed:

“rs147239461 at *IGF2/H19(A)* is located downstream from H19 35 Kb from the H19-ICR. Therefore, it is unlikely that the variant directly disrupts methylation of the ICR. However, since the variant and three of its strong correlates are located at a CTCF binding site it might

affect methylation at the ICR and thus disturb the balanced expression of imprinted genes in the region.”

For the remaining variants, rs143840904 at *KCNQ1* and rs41286560 in *RTL1*, we have already attempted a mechanistic explanation that reads as follows:

- For rs143840904 located at *KCNQ1* on page 8, paragraph 1;

“In contrast to the situation in Sardinia, rs143840904 has no strong correlates in Iceland (all $r^2 < 0.73$) (Fig. 2 and Supplementary Fig. 5), and conditional analysis revealed that it alone accounts for the maternal effect ($P_{adj} > 0.05$ for each of the other five variants, Supplementary Table 11), indicating that rs143840904 is the causative variant. Rs143840904 is located in a EZH2 transcription factor binding site (Supplementary Table 11). Interestingly, EZH2 is the functional enzymatic component of polycomb repressive complex 2 (PRC2), which along with the paternally expressed long noncoding RNA *KCNQ1OT1* participates in maintaining monoallelic expression at the *KCNQ1* imprinted domain”

- For rs41286560 in *RTL1* on page 12, paragraph 2;

“We note that the height-associated missense variant (rs41286560) with parent-of-origin effect is located 42 bp from miR127 that has been linked to placentomegaly in mice when maternally knocked out. Given that the effect is predominantly from the paternal chromosome we find it more likely that the effect is mediated through missense in *RTL1* itself. Paternal *Rtl1* knockout mice develop growth retardation at the fetal stage that persists into adulthood, but to our knowledge *RTL1* has not been linked to growth defects in humans.”

Minor comments:

Comment 5: Line 265 (page 12):

Conclusions about no association with birth weight are not justified.

First, birth weight is reported with much more error, second the sample size for BW was much smaller. Therefore I don't think that any conclusion can be made with the available data.

Answer:

To answer the reviewers comment we have added confidence intervals for adult height and birth length effects of the 64 independent signals in Supplementary Table 4 (excel supplement).

Furthermore, we have modified the text, and qualified the conclusion, regarding associations with birth length (underlined text):

“First, for the variants with parent-of-origin effects on adult height, one is associated with birth length while three are not, which might suggest that they affect growth after weaning.” (page 13, paragraph 1)

Comment 6: Line 266-269 (page 12): The expected paternal height increase and maternal height decreasing effects are not easy to check here: Depends which allele one looks at. If the baseline is the homozygous minor group, then the effects look opposite. If e.g. for rs147239461 we take the reference the TT group then the effect of mat-G is negative and pat-G is positive , but if the reference is the GG group then mat-T is positive and pat-T is negative then, one can argue both ways.

Answer: In the parent-of-origin GWAS the maternal and paternal alleles are tested separately and the effect is always reported for the minor allele. When inherited from the father the minor allele of rs147239461 has a decreasing effect and when inherited from the mother the minor allele has an increasing effect on height. When inherited from both parents the effects mask out each other (see Fig. 3). The exact opposite applies if the major allele is the effect allele. In the main text we refer to the variant as rs147239461[T] to clarify that the minor allele is the effect allele.

Comment 7: Line 425 (page 19): "had a more significant association with height, in our data than their reported correlates" This can be random fluctuation; this is not a solid reason to say it is a refined signal.

Answer:

In addition to reaching the threshold of genome wide significance, we state that for a signal to be classified as a refinement of previously reported signal it has to fulfill all four of the following criteria (see methods):

- (i) the refinement variant is moderately correlated with at least one reported variants ($0.10 < r^2 < 0.80$)
- (ii) the refinement variant demonstrates a significantly stronger association with height in our data than its reported correlates
- (iii) that the refinement variant accounts for the effects of its reported correlates (based on conditional analysis)
- (iv) the reported correlates could not fully account for the effect of the refinement variant detected in the present study (adjusted $P < 1.0 \times 10^{-3}$)(based on conditional analysis).

The results for the conditional analysis of the refinement variants are in Supplementary Table 6 (excel sheet).

To clarify that we required all criteria to be fulfilled, we simplified the text and made the following modifications (underlined text) (page 20, paragraph 3):

“Variants were classified as a refinement of a formerly reported signal if they fulfilled all the following three criteria:

- (i) the refinement variant is moderately correlated with at least one reported variants ($0.10 < r^2 < 0.80$)
- (ii) the refinement variant demonstrates a significantly stronger association and accounts for the effects of its reported correlates (based on conditional analysis)
- (iii) the reported correlates could not fully account for the effect of the refinement variant detected in the present study (adjusted $P < 1.0 \times 10^{-3}$)(based on conditional analysis).“

12 signals fulfill all three criteria. If criteria (iii) is changed to $P < 1.0 \times 10^{-4}$ then 8 remain and if changed to $P < 1.0 \times 10^{-5}$, 7 remain.

Reviewer #2 (Remarks to the Author):

This is a compelling story of the identification of several new variants associated with human growth, with a particular focus on four, 3 of which are novel, that do so with parent of origin effects. The power of the Decode study and its ability to impute parental genotypes and assess the effects of non transmitted as well as transmitted alleles makes the study unique. The parent of origin effects are very likely to be due to imprinting. The results build a fascinating story around the variation in the IGF2/H19 region in particular. The added interest is that the maternal effects appear to increase growth, whilst paternal effects decrease growth, contrary to the evolutionary conflict theory that never had too much evidence.

I have only minor suggestions for the authors to consider, but it looks like a well polished manuscript:

Comment 1. It would be worth stating in a few more places what the effects are in cm as well as SDs. As height is very close to normally distributed, then a simple conversion should suffice - some of the additive and recessive effects of low frequency variants look like v large effects.

We agree with the reviewer that adding height effects in cm will be more informative for evaluation of the effects of the reported variants on height. In the revised manuscript, we have added height effects in cm in the following places:

Page 6, paragraph 4

Page 7, paragraph 2

Page 7, paragraph 3

Page 8, paragraph 2

Page 10, paragraph 2

To make it clear to what the height effect in cm corresponds to we have added the following sentence on page 5, paragraph 1:

“The overall standard deviation used in the paper is 6.6.cm.”

Comment 2. The conclusion that "take together our results demonstrate that genomic imprinting affects variation of anthropometric traits in humans" should be qualified because we knew this. It is that the results show multiple that more common variation , especially in the IGF2/H19 region affect human growth by parental imprinting "?

We agree with the reviewer and have modified the concluding remarks accordingly.

We wrote (page 13, paragraph 2):

"Taken together our results demonstrate that genomic imprinting affects variation of anthropometric traits in humans"

This has been changed to:

"Taken together our results demonstrate that common variations affects human growth by parental imprinting in humans"

Reviewer #1 (Remarks to the Author)

The authors answered my comments appropriately and executed modifications to the m/s to improve it substantially. Only a few outstanding points remained:

1. My question remained unanswered: "How do the samples/batches map on the genetic PCA map?" May I see this plot?

2. "Our method of testing for association takes the closest relatedness into account using a mixed effect model, but is not able to fully account for the relatedness between individuals." I'm not sure I understand this, mixed models use the genetic kinship matrix estimated from the genetic data. Did the authors use only the genealogical information to build the kinship matrix? If so, why not the genetic – it is supposed to be less prone to errors and ensure better stratification correction.

3. The reported new additive SNVs can be interesting, but if the authors insist on adding these results would be more convincing if at least an attempt were made to replicate it in the recently submitted GIANT height exome-chip study. (Reverse replication was already done.)

4. "First, for the variants with parent-of-origin effects on adult height, one is associated with birth length while three are not, which might suggest that they affect growth after weaning." – I still do not see it justified (even if the word might added). For the POE height replication in other European cohorts the authors (correctly) claim that it would be underpowered so they do not even attempt it. While this birth weight POE analysis is equally underpowered (only 12K samples with a much noisier outcome than height), they do the analysis and draw conclusions.

5. Regarding the minor allele being the effect allele: It is of course fine with me, the authors may chose whichever allele they wish to call it effect allele. My problem is the "apparent" contradiction with the paternal conflict hypothesis (Page 13, lines 292-295). We can turn the sentence to "the paternally inherited major alleles at IGF2 and RTL1 result in increased adult height, and in the case of rs147239461 at IGF2/H19 increased birth length" Reading this way there is no contradiction. I see nothing special about the minor allele.

Reviewer #2 (Remarks to the Author)

no further comments

Epigenetic and genetic components of height regulation

by Benonisdottir et al.

Reviewer #1 (Remarks to the Author):

The authors answered my comments appropriately and executed modifications to the m/s to improve it substantially.

Only a few outstanding points remained:

1. My question remained unanswered: “How do the samples/batches map on the genetic PCA map?” May I see this plot?

To address the reviewers request, we have provided principle component analysis (PCA) plots (see Figure 1 below) for 150 thousand Icelanders typed using eleven different Illumnia SNP chips. Three of them are human hap chips (a, b, c) and eight of them are omni chips (d, e, f, g, h, i, j, k). We generated PCA plots depicting the first two principal components of chip-typed individuals for each SNP chip. The type of SNP chip used explains 0.46% of the variance of the first principal component (PC1), 0.28 % of PC2 and less then 0.1% for all other PC.

Figure 1: PCA plots for 150 thousand individuals typed on three different human hap chips (a, b, c) and eight different omni chips (d, e, f, g, h, i, j, k).

Chip-type explains 0.0007% of the variance in adult height in our dataset (N=80,546) as assessed by linear regression between chip-type and adult height (adjusted for age and sex). To further examine the effect of chip-type we ran a case-control analysis (omni vs. human hap) and tested if the 64 variants, reported in the present study to associate with adult height, associated with chip-type ($N_{\text{omni}} = 100,201$; $N_{\text{human-hap}} = 34,329$). None of the 64 variants associated with chip-type under the additive model at a significance threshold accounting for multiple testing, $P = 0.05/64$.

- 2. “Our method of testing for association takes the closest relatedness into account using a mixed effect model, but is not able to fully account for the relatedness between individuals.” I’m not sure I understand this, mixed models use the genetic kinship matrix estimated from the genetic data. Did the authors use only the genealogical information to build the kinship matrix? If so, why not the genetic – it is supposed to be less prone to errors and ensure better stratification correction.**

The reviewer has raised a valid point. While it is right that using genetic information is less prone to errors, due to the large size of our data (> 150,000) it is not computationally feasible to implement this method. Until now we have used genealogical information to build a kinship matrix. However, recently a more efficient mixed-model method has been developed, BOLT-LMM, which is far less demanding computationally and will allow us to make use of a genetic kinship matrix based on genetic data (Loh et al., PMID:25642633). Our future work will implement this method when testing for associations.

- 3. The reported new additive SNVs can be interesting, but if the authors insist on adding these results would be more convincing if at least an attempt were made to replicate it in the recently submitted GIANT height exome-chip study. (Reverse replication was already done.)**

There are at least two reasons why an attempt to replicate our new additive association signals using „the recently submitted GIANT height exome-chip study“ is inadvisable. First, that study is not yet published. Second, seven of the nine new additive association signals involve variants with a minor allele frequency (MAF) lower than 1% (thereof five lower than 0.7%). In the manuscript, we used the Exac and 1000 genomes data to show that these variants are either absent or vanishingly rare in populations outside Iceland (see Supplementary Table 15). This severely limits the possibility of replication in studies such as the GIANT height exome-chip study. The two more common additive variants (MAF=2.40% and 13.86%) have very convincing P -values (both $\leq 10^{-14}$), and we explain in our Supplementary Text why previous GWAS with common variants did not detect these associations (Supplementary Text, page 4, paragraph 4 and page 5, paragraph 1-2)

In the Supplementary Text, we present strong supporting evidence for each of the nine novel additive association signals. Three of the rare variants (*NPR2*, *ACAN*, *GH1*) are in genes with other mutations that are known to cause monogenic growth defects according to OMIM (see Supplementary table 14). Two of the variants are in the *ZFAT* gene, that harbours a well replicated but uncorrelated common variant associated with adult height (Supplementary text, page 3, paragraph 2). The variant in *TET1* is associated with both adult height and birth length in our data and *Tet1* paternal knockout mice have been shown to exhibit decreased growth. Two of the variants are in the genes *ADAMTS10* and *ADAMTS17*, which harbour rare mutations known to cause monogenic growth defects according to OMIM. The mutation Gln256ProfsTer38 in *ADAMTSL4* has been reported to cause ectopia lentis (connective tissue disorder) under a recessive mode of inheritance (see Supplementary Text, page 4, paragraph 3) – we now demonstrate that it also has an additive effect on height.

In short, we believe these additive association results to be robust and important. They not only warrant incorporation in our manuscript for that reason, but also because they derive from the study design that was used to detect the parent of origin variants.

4. **“First, for the variants with parent-of-origin effects on adult height, one is associated with birth length while three are not, which might suggest that they affect growth after weaning.” – I still do not see it justified (even if the word might added). For the POE height replication in other European cohorts the authors (correctly) claim that it would be underpowered so they do not even attempt it. While this birth weight POE analysis is equally underpowered (only 12K samples with a much noisier outcome than height), they do the analysis and draw conclusions.**

To address the reviewers concerns we have removed this sentence from the manuscript (Page 13, paragraph 1).

5. **Regarding the minor allele being the effect allele: It is of course fine with me, the authors may chose whichever allele they wish to call it effect allele. My problem is the “apparent” contradiction with the paternal conflict hypothesis (Page 13, lines 292-295). We can turn the sentence to “the paternally inherited major alleles at IGF2 and RTL1 result in increased adult height, and in the case of rs147239461 at IGF2/H19 increased birth length” Reading this way there is no contradiction. I see nothing special about the minor allele.**

We respectfully disagree with the reviewer on this point. As can be seen in Fig 3, our original sentence is correct in four of four instances for the two IGF2/H19 variants and the RTL1 variant – i.e. mean adult height (and birth length in the case of rs147239461) is reduced when the minor allele is inherited from the father. However, the reviewer’s rewrite of the sentence is only correct for two of four instances (rs147239461 and rs7482510 with adult height) – and in both cases the reduction in height from the paternal minor allele is more marked than the increase in height from the paternal major allele.

Another point to bear in mind is that in each case the minor allele is the derived (i.e. more recent) allele. Thus, it follows that the introduction of variation at these loci (i.e. the mutations that gave rise to the derived/minor alleles) resulted in reduced rather than increased height in the population. We therefore left the sentence in question unchanged.

Reviewers' Comments:

Reviewer #1 (Remarks to the Author)

All my comments have been reassuringly addressed.